# Stimulation of Distinct Rhizosphere Bacteria Drives Phosphorus and Nitrogen Mineralization in Oilseed Rape under Field Conditions

Ian D. E. A. Lidbury,[a,b] Sebastien Raguideau,[a] Chiara Borsetto,[a] Andrew R. J. Murphy,[a] Andrew Bottrill,[a] Senlin Liu,[a,c] Richard Stark,[a] Tandra Fraser,[d] Andrew Goodall,[d] Alex Jones,[a] Gary D. Bending,[a] Mark Tibbet,[d] John P. Hammond,[d] Chris Quince,[e] David J. Scanlan,[a] Jagroop Pandhal,[f] Elizabeth M. H. Wellington[a]

aSchool of Life Sciences, University of Warwick, Coventry, United Kingdom
bPlants, Photosynthesis, and Soil, School of Biosciences, University of Sheffield, Sheffield, United Kingdom
cKey Laboratory of Agricultural Environmental Microbiology, Ministry of Agriculture and Rural Affairs, College of Life Sciences, Nanjing Agricultural University, Nanjing, Jiangsu, People's Republic of China
dSchool of Agriculture, Policy, and Development, University of Reading, Earley Gate, Whiteknights, Reading, United Kingdom
eEarlham Institute, Norwich Research Park, Norwich, United Kingdom
fDepartment of Chemical and Biological Engineering, University of Sheffield, Sheffield, United Kingdom

**ABSTRACT** Advances in DNA sequencing technologies have drastically changed our perception of the structure and complexity of the plant microbiome. By comparison, our ability to accurately identify the metabolically active fraction of soil microbiota and its specific functional role in augmenting plant health is relatively limited. Important ecological interactions being performed by microbes can be investigated by analyzing the extracellular protein fraction. Here, we combined a unique protein extraction method and an iterative bioinformatics pipeline to capture and identify extracellular proteins (metaexoproteomics) synthesized in the rhizosphere of *Brassica* spp. We first validated our method in the laboratory by successfully identifying proteins related to a host plant (*Brassica rapa*) and its bacterial inoculant, *Pseudomonas putida* BIRD-1. This identified numerous rhizosphere specific proteins linked to the acquisition of plant-derived nutrients in *P. putida*. Next, we analyzed natural field-soil microbial communities associated with *Brassica napus* L. (oilseed rape). By combining metagenomics with metaexoproteomics, 1,885 plant, insect, and microbial proteins were identified across bulk and rhizosphere samples. Metaexoproteomics identified a significant shift in the metabolically active fraction of the soil microbiota responding to the presence of *B. napus* roots that was not apparent in the composition of the total microbial community (metagenome). This included stimulation of rhizosphere-specialized bacteria, such as *Gammaproteobacteria*, *Betaproteobacteria*, and *Flavobacteriia*, and the upregulation of plant beneficial functions related to phosphorus and nitrogen mineralization. Our metaproteomic assessment of the "active" plant microbiome at the field-scale demonstrates the importance of moving beyond metagenomics to determine ecologically important plant-microbe interactions underpinning plant health.

**IMPORTANCE** Plant-microbe interactions are critical to ecosystem function and crop production. While significant advances have been made toward understanding the structure of the plant microbiome, learning about its full functional role is still in its infancy. This is primarily due to an incomplete ability to determine *in situ* plant-microbe interactions actively operating under field conditions. Proteins are the functional entities of the cell. Therefore, their identification and relative quantification within a microbial community provide the best proxy for which microbes are the most metabolically active and which are driving important plant-microbe interactions. Here, we provide the first metaexoproteomics assessment of the plant microbiome using field-grown oilseed rape as

**Ad Hoc Peer Reviewer** Jean Armengaud, CEA

Address correspondence to Ian D. E. A. Lidbury, l.lidbury@sheffield.ac.uk, or Elizabeth M. H. Wellington, E.M.H.Wellington@warwick.ac.uk.

The authors declare no conflict of interest.

the model crop species, identifying key taxa responsible for specific ecological interactions. Gaining a mechanistic understanding of the plant microbiome is central to developing engineered plant microbiomes to improve sustainable agricultural approaches and reduce our reliance on nonrenewable resources.

**KEYWORDS** *Brassica napus*, field soil, metagenomics, metaproteomics, plant microbiome, sustainable agriculture

The plant microbiome is integral to plant health as it delivers several life support functions (1, 2). This includes enhancing the plant's ability to acquire both macro- and micronutrients, such as nitrogen, phosphorus, and iron, as well as enhancing plant innate immunity against a range of plant pathogens (2–4). Since the green revolution, intensive agricultural practices have resulted in a decoupling between microbes and their host plants (5). The breakdown of rhizobia-legume symbiosis in heavily fertilized cropping systems is perhaps the most well-known example (6). Others, such as the apparent reduction in the relative abundance of *Bacteroidetes* in domesticated crops relative to their wild cultivars, are more cryptic (7). Agriculture is now facing a significant global crisis: a rapidly changing climate, an ever-growing human population, and depletion of our natural resources used to fuel crop production has identified severe vulnerabilities in ensuring future food security (2, 8). While the industrial production of nitrogen fertilizers is a highly energetic process, the production of inorganic phosphorus fertilizers is reliant on the continued supply of mined rock phosphate (9). The latter of these fertilizer production regimes is set to cause various socioeconomic and political tensions as global stocks of rock phosphate are depleted (9, 10). Thus, there is an urgent need to develop a holistic understanding of the plant microbiome function and its numerous components (11).

Through the release of signaling molecules, exudation of organic nutrients, and the decoration of plant cell walls with specific attachment molecules, plants can actively select for a subset of specialized soil microorganisms (12, 13). This frequently involves a reduction in microbial diversity as one moves from the bulk soil > rhizosphere > root tissue (1, 14). While bulk soil is considered a relatively carbon poor environment favoring an oligotrophic lifestyle, the rhizosphere and root system is typified by a high turnover of organic matter driven through rhizodeposition, an environment favoring a copiotrophic lifestyle. Indeed, copiotrophic bacteria related to *Proteobacteria*, *Bacteroidetes*, and *Actinobacteria* often dominate plant-associated microbial communities (15, 16).

While our understanding of the diversity, structure, and functional potential of microbial communities has drastically improved, there is still considerable uncertainty about how this translates into specific plant-microbe interactions, especially carbon for nutrient exchange (2). Therefore, we still lack understanding of the functional components involved in delivering beneficial plant activities within the root microbiome. Proteins are the functional entities of the cell whose regulation is controlled by surrounding biotic and abiotic conditions. Metaproteomics, the study of the entire protein content of a given environmental sample, holds enormous potential to improve our understanding regarding the function of soil microbial communities (17). Unlike its application in seawater (18, 19), anaerobic digestors (20, 21), or the human or animal gut (22, 23), soil metaproteomics has been relatively underexploited (24). This is partly due to conventional soil extraction methods coextracting contaminating substances, such as organic carbon and humic acids (24). Furthermore, microbial complexity in soils is usually greater than any other environment (1, 11, 24), leading to considerable problems in metagenome sequencing and assembly, which are critical for quality metaproteome measurements. However, its application is increasing due to improved bioinformatics pipelines to correctly identify peptides from mass-spec data sets (25). Furthermore, the majority of expressed proteins are related to cytoplasmic housekeeping and core metabolic functions, which can often result in poor detection of more ecologically important but less abundant noncytoplasmic proteins (26). This observation is

evident in our previous laboratory-based studies investigating individual bacterial responses to phosphate limitation (27, 28). One alternative is to focus on the extracellular (exo) fractions of proteins found outside the cell using metaexoproteomics (MEP), a method that adapts extraction protocols for detecting soil extracellular enzyme activity (29). MEP has been successfully utilized to determine the active chitin-degrading community of a tropical soil in response to chitin amendment (29). While this extraction method is applicable to bulk soil analysis (requiring 50 to 100 g soil material), sampling the rhizosphere (typically 1 to 2 g material) is much more challenging. Furthermore, the method currently requires specialized equipment and is relatively low throughput. These technical limitations have likely reduced the take up of this approach, despite its enormous potential.

Our recent work has successfully characterized the *in vitro* exoproteomes of single strain cultures related to *Pseudomonas* spp. and *Flavobacterium* spp. in response to phosphate-limiting growth conditions (27, 28). These rhizobacteria produce numerous hydrolytic and transport proteins targeting organic phosphorus components in response to phosphate limitation. Thus, exoproteomics can generate significant insights into the mechanisms utilized by microbes to compete for growth limiting nutrients and their contribution to environmental nutrient cycling (26, 30, 31). In this study, we adapted our previous extraction method to efficiently capture the extracellular proteins (metaexoproteome [MEP]) found in agricultural soils to identify the most active microbial taxa in the rhizosphere of *Brassica napus* L. (oilseed rape) and the major metabolic interactions operating. We hypothesized that (i) the rhizosphere would contain a distinct set of metabolically active microbes relative to the surrounding bulk soil and (ii) microbes would express proteins for the mineralization of N and P as a response to elevated C. In addition to capturing extracellular plant and aphid-pest proteins in the rhizosphere, we observed a distinct shift in active fraction of the microbial community in this compartment relative to the bulk soil with several *Pseudomonas* spp. dominating the MEP.

## RESULTS

***Pseudomonas putida* BIRD-1 expresses a distinct set of rhizosphere-associated proteins under laboratory conditions.** Our previous extraction method involved extracting protein from ~100 g soil (29), which is not feasible when working with rhizosphere soil. To determine whether we could efficiently capture extracellular proteins from soil using the Stratabead resin, we first performed a series of protein spikes into soil and water as a control. We used either bovine serum albumin or the exoproteome of lab-grown *P. putida* BIRD-1 as the protein spike, both of which were successfully recaptured from soil and water (Fig. S2).

To further confirm the efficacy of this method, we established a simple plant growth experiment under laboratory conditions using *Brassica rapa* OH17 grown in a sand: soil mix ($n = 6$) and inoculating washed and resuspended *Pseudomonas putida* BIRD-1 ($6 \times 10^8$ cells mL$^{-1}$). After 3 weeks of growth, the sand: soil mix was collected, and protein was extracted. Using the *in silico* predicted proteome of either *P. putida* BIRD-1 or *B. napus* as the database and a criterion of at least 2 unique peptides per protein, we identified a total 201 (177 protein clusters) and 215 proteins, respectively (Table S1a and b). We also performed a third search using the *P. putida* BIRD-1 proteome and additional protein sequences related to various extracellular proteins (substrate binding proteins, phosphatases, chitinases, pectinases, etc.) retrieved from bacterial genomes deposited in the IMG/JGI database (accessed 05/06/2018). This search indicated that both *P. putida* BIRD-1 proteins, in addition to those from closely related *Pseudomonas* strains, and exoproteins produced by taxonomically divergent rhizobacteria were captured by our method (Table S1c). Two replicate samples containing low quantities of protein were omitted from further analyses. While normalized spectral abundance factor (NSAF) quantification methods have now been superseded by various peak areas methods, we used NSAF here to make easier comparisons with our previously published data sets (27). For *P. putida* BIRD-1, the 40 most abundant proteins represented 61% of the total exoproteome. The pot-grown *P. putida* exoproteome (black bar) showed a distinct profile compared to previous *in vitro* exoproteomes (28) (blue bar, Fig. 1A and Table S1b). This included an increase

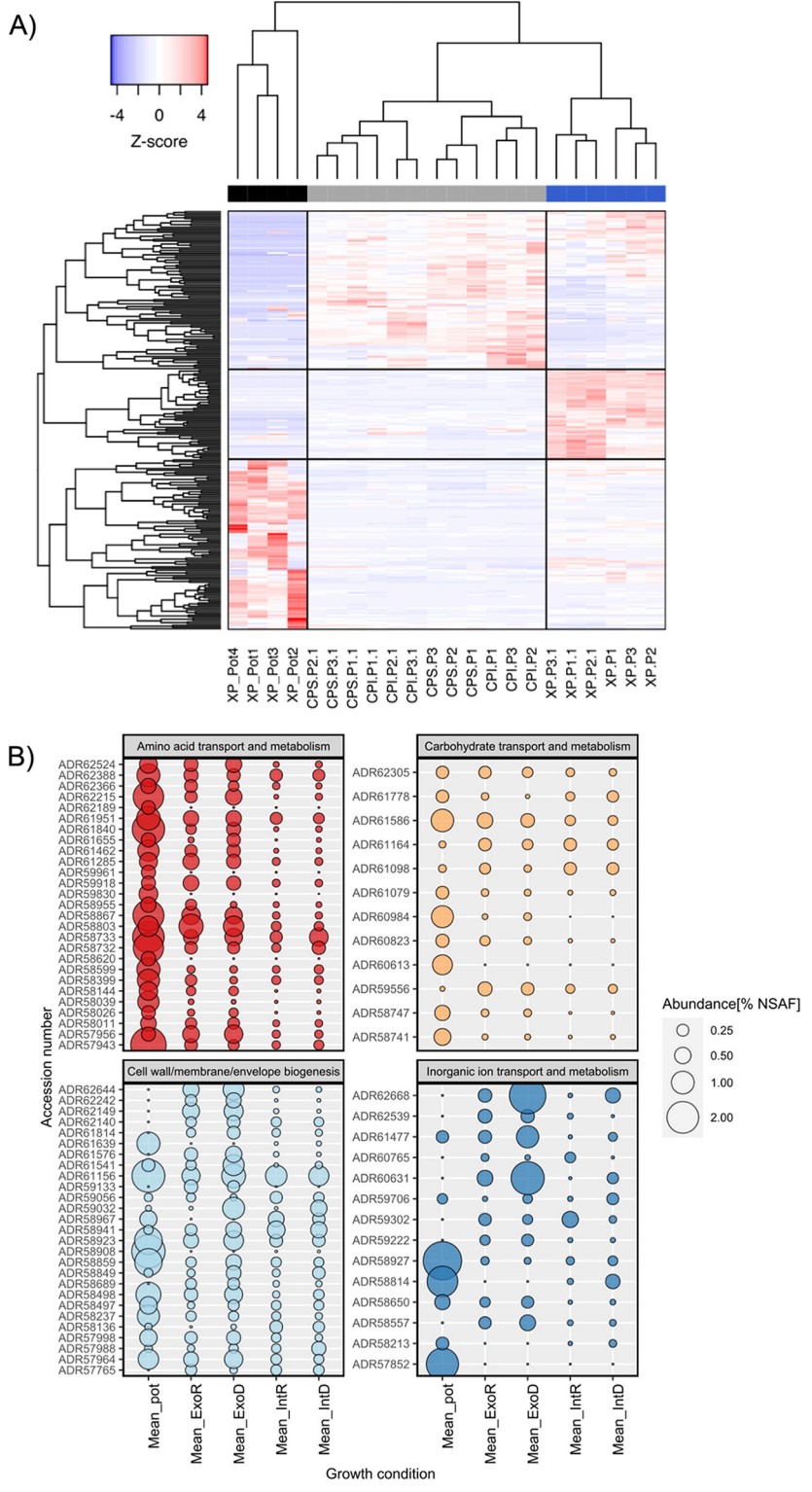

**FIG 1** Comparison of *P. putida* BIRD proteome during *in situ* and *in vitro* growth experiments. (A) Hierarchical clustering of proteins based on abundance profiles (Z-score, calculated from % abundance) across the different growth conditions; blue, exoproteome (XP) of liquid cultures; gray, insoluble and soluble fractions of the cellular proteome (CP) from liquid cultures; black, exoproteome captured from pot experiments using *Brassica rapa* R018. Data for individual replicates are displayed. (B) Assessment of functions (COG categories) related to periplasmic, cell surface, and extracellular proteins across the different growth conditions. The mean value for quadruplicate (pot), triplicate (*In vitro* deplete/replete), and sextuple (*In vitro* deplete/replate) replicates from each growth conditions is plotted. GenBank accession number are given on the *y* axis. All % abundance values were calculated from normalized Spectral Abundance Factor (NSAF) values.

in the relative abundance of some outer membrane proteins, porins and substrate binding proteins associated with ABC transport systems and a decrease in others, while some were detected at similar relative abundances (Table 1). While we did detect several cytoplasmic proteins, their relative abundance in either the exo- or intracellular soluble (gray bar) proteomes were still markedly different. In addition, many of the abundant *in vitro* intracellular proteins detected in our previous study (28) were either not detected or were present in very low abundance in the pot-grown proteome suggesting an inherent change in metabolism. Most of the abundant rhizosphere-inducible proteins belonged to four general Clusters of Orthologous Gene (COG) categories: (i) amino acid transport and metabolism, (ii) carbohydrate transport and metabolism, (iii) inorganic ion transport and metabolism, and (iv) cell envelope biogenesis, outer membrane (Fig. 1B and Table 1). While several substrate binding proteins related to carbon and nitrogen metabolism were enriched in the rhizosphere (Fig. 1B and Table S1b), substrate binding proteins associated with high-affinity phosphate (PstS) or 2-aminophosphonate (AepX) ABC transporters were not detected, nor was the low-phosphate inducible alkaline phosphatase PhoX, that were all detected *in vitro* (27, 32). This would suggest that under these conditions BIRD-1 did not experience localized phosphate depletion severe enough to trigger its P-stress response regulon (28).

**Metaproteomic assessment of microbial activity in the rhizosphere of field-grown oilseed rape and surrounding bulk soil.** Next, we sampled bulk and rhizosphere soil from agricultural fields sown with oilseed rape under contrasting P fertilizer regimes (Fig. S1). Plants were sampled at an early growth stage, between the four-six leaf and rosette growth stages. To create a comprehensive database for MEP, we generated over 500 GB of metagenomics data from bulk and rhizosphere soil samples and used a coassembly method to identify over 64 million open reading frames (ORFs). When controlling for one of the two variables separately, nonmetric multidimensional scaling (NMDS) analysis showed no significant effect of compartments (bulk versus rhizosphere; one-way similitude analysis [ANOSIM]: $R = 0.027$, $P = 0.316$) or fertilizer treatment (High P versus Low P; ANOSIM: $R = 0.485$, $P = 0.11$) on the total soil microbial communities (metagenome [MG]) (Fig. 2A and B). Again, using a minimum of two unique peptides per protein, a total of 1,895 (10 comtaminants) (Table S2c) proteins were detected across all samples. The relative abundance of proteins was quantified using label-free quantification (LFQ) values. There was a significant difference (Adonis: $R^2 = 0.736$, $P = 0.001$) in the proteomic profiles of bulk and rhizosphere samples (Fig. 2C). In contrast, there was no significant effect (Adonis: $R^2 = 0.031$, $P = 0.667$) of fertilizer treatment on the proteomic profiles of either soil compartment. While only 48 proteins were significantly enriched in the bulk soil (false-discovery rate [FDR] corrected: $P < 0.05$; fold change: >1.5), 815 were significantly enriched (FDR corrected: $P < 0.05$; fold change: >2) in the rhizosphere (Table S2c). Furthermore, almost all highly abundant proteins were rhizosphere associated, with only a few abundant proteins associated with bulk soil (Fig. 2D).

**Autochthonous *Pseudomonas* spp. were highly active in the rhizosphere of young field-grown *Brassica napus* L.** The vast majority of rhizosphere protein content was related to several *Pseudomonas* spp. (~65%) and *B. napus* (~20%) (Fig. 3A). In addition, several abundant rhizosphere proteins were related to soil/root aphids and its corresponding symbiont *Buchnera aphicola* (*Gammaproteobacteria*). Proteins expressed by these groups as well as *Betaproteobacteria*, other *Gammaproteobacteria*, *Bacteroidetes* (predominantly *Flavobacteraceae*), and fungi were all rhizosphere-enriched (Fig. 3A). In contrast, proteins expressed by *Actinobacteria*, *Alphaprotebacteria*, *Acidobacteria*, and *Archaea* were collectively more abundant in bulk soil (Fig. 3A). The identified *Pseudomonas* proteins were aligned to four different *Pseudomonas* genomes, representing four distinct *Pseudomonas* groups (33): *P. putida* BIRD-1, *P. fluorescens* SBW25, *P. stutzeri* DSM4166, and *P. syringae* DC3000. On average, detected proteins had the highest identity with *P. fluorescens* SBW25 (93%), suggesting most of the identified proteins belonged to strains within the *P. fluorescens* group. However, there was significant variation in average identity (%) related to each strain and numerous proteins were absent from each individual genome (Fig. S3 and

**TABLE 1** Comparison of extracellular proteins with validated/predicted outer membrane or periplasmic localization detected in the exoproteome of *P. putida* BIRD-1 grown in *B. rapa* R-o-18 rhizosphere soil (Mean_pot) *in vitro* (phosphate replete [Mean_ExoR]) and phosphate deplete [ExoD] growth conditions)

| Accession | Gene description | General COG | % abundance (based on NSAF)[a] | | |
| --- | --- | --- | --- | --- | --- |
| | | | Rhizo exo | In vitro ExoR | In vitro ExoD |
| **Outer membrane proteins** | | | | | |
| ADR58908 | OprH: outer membrane protein | Cell wall/membrane/envelope biogenesis | 2.220 | 0.002 | 0.002 |
| ADR61156 | OprF: outer membrane porin | Cell wall/membrane/envelope biogenesis | 2.155 | 0.623 | 1.104 |
| ADR58923 | OprD: outer membrane porin | Cell wall/membrane/envelope biogenesis | 1.508 | 0.414 | 0.743 |
| ADR58859 | 17-kDa surface antigen | Cell wall/membrane/envelope biogenesis | 1.467 | 0.192 | 0.450 |
| ADR58498 | Hypothetical outer membrane protein, conserved | Cell wall/membrane/envelope biogenesis | 1.213 | 0.328 | 0.617 |
| ADR61639 | OmpA/MotB domain protein | Cell wall/membrane/envelope biogenesis | 1.026 | 0.002 | 0.002 |
| ADR58237 | OprG: outer membrane porin | Cell wall/membrane/envelope biogenesis | 0.948 | 0.182 | 0.175 |
| ADR57964 | OprD: outer membrane porin | Cell wall/membrane/envelope biogenesis | 0.797 | 0.397 | 0.644 |
| ADR58967 | Hypothetical protein, conserved | Cell wall/membrane/envelope biogenesis | 0.550 | 0.002 | 0.002 |
| ADR61541 | Hypothetical protein, conserved | Cell wall/membrane/envelope biogenesis | 0.302 | 0.188 | 0.859 |
| ADR58941 | OprL: outer membrane porin | Cell wall/membrane/envelope biogenesis | 0.148 | 0.247 | 0.477 |
| ADR58485 | Putative outer membrane lipoprotein | Cell wall/membrane/envelope biogenesis | 0.055 | 0.349 | 0.378 |
| ADR58849 | OmpA-like: outer membrane porin | Cell wall/membrane/envelope biogenesis | 0.137 | 0.108 | 0.398 |
| ADR61817 | OsmY-like: transport-associated protein | Cell wall/membrane/envelope biogenesis | 0.002 | 0.602 | 0.695 |
| ADR59032 | Outer membrane lipoprotein, conserved: phosphate-repressible | Cell wall/membrane/envelope biogenesis | 0.052 | 0.020 | 0.929 |
| ADR62644 | Outer membrane lipoprotein, conserved | Cell wall/membrane/envelope biogenesis | 0.002 | 0.494 | 0.847 |
| ADR58940 | TolB: periplasmic component of the Tol biopolymer transport system | Cell wall/membrane/envelope biogenesis | 0.002 | 0.907 | 1.598 |
| ADR62149 | Rare lipoprotein A family | Cell wall/membrane/envelope biogenesis | 0.002 | 0.544 | 0.702 |
| ADR59302 | FpvA: TonB-dependent iron receptor | Cell wall/membrane/envelope biogenesis | 0.002 | 0.276 | 0.169 |
| ADR58775 | PhoX: alkaline phosphatase | Cell wall/membrane/envelope biogenesis | 0.002 | 0.031 | 1.149 |
| **Periplasmic substrate binding proteins** | | | | | |
| ADR61840 | AapJ: amino acid ABC transporter substrate-binding protein, PAAT family | Amino acid transport and metabolism | 2.087 | 0.193 | 0.361 |
| ADR58867 | BraC: L-leucine-binding protein/L-isoleucine-binding protein/L-valine-binding protein | Amino acid transport and metabolism | 1.911 | 0.653 | 0.570 |
| ADR62215 | LivJ: amino acid/amide ABC transporter substrate-binding protein, HAAT family | Amino acid transport and metabolism | 1.841 | 0.271 | 0.521 |
| ADR61462 | Substrate-binding region of ABC-type glycine betaine transport system | Amino acid transport and metabolism | 0.880 | 0.271 | 0.326 |
| ADR58803 | L-glutamate-binding protein/L-aspartate-binding protein | Amino acid transport and metabolism | 0.819 | 1.148 | 0.812 |
| ADR58144 | Polyamine ABC transporter, periplasmic polyamine-binding protein | Amino acid transport and metabolism | 0.653 | 0.154 | 0.155 |
| ADR62524 | PotF_2: putrescine ABC transporter, periplasmic putrescine-binding protein | Amino acid transport and metabolism | 0.601 | 0.357 | 0.494 |
| ADR57956 | L-cystine-binding protein/diaminopimelate-binding protein | Amino acid transport and metabolism | 0.356 | 0.531 | 0.646 |
| ADR62668 | PstS1: phosphate ABC transporter, periplasmic phosphate-binding protein | Inorganic ion transport and metabolism | 0.002 | 0.328 | 2.680 |
| ADR60631 | PstS2: phosphate ABC transporter, periplasmic phosphate-binding protein | Inorganic ion transport and metabolism | 0.002 | 0.453 | 2.198 |
| ADR61477 | AepS: phosphonate/ABC-type Fe3+ transport system periplasmic component-like protein | Inorganic ion transport and metabolism | 0.002 | 0.266 | 0.467 |
| ADR62476 | AepX: 2-aminoethylphosphonate ABC transporter, periplasmic binding protein | Inorganic ion transport and metabolism | 0.932 | 0.046 | 0.097 |
| ADR58557 | Phosphonate/selenate ABC transporter periplasmic phosphonate-binding protein | Inorganic ion transport and metabolism | 0.270 | 0.309 | 0.959 |
| ADR60984 | Carbohydrate ABC transporter substrate-binding protein, CUT1 family | Carbohydrate transport and metabolism | 0.002 | 0.002 | 0.053 |

[a]Values present are the calculated mean % abundance ($n = 4$) in the total exoproteome based on normalized spectral abundance factor (NSAF) values.

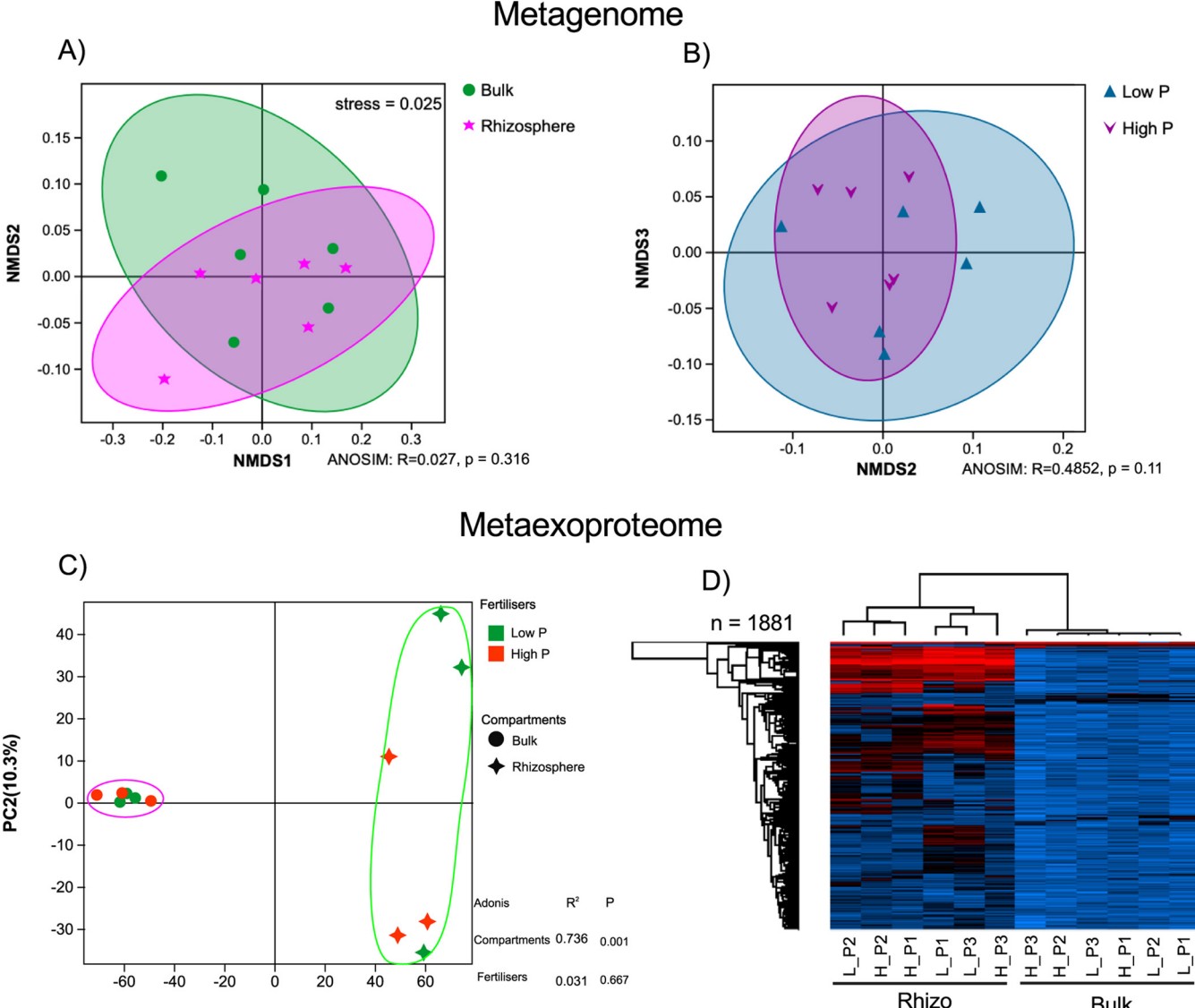

**FIG 2** Metagenomic and metaexoproteomic assessment of field-grown *Brassica napus* L. bulk soil and rhizosphere communities. (A and B) NMDS ordination (stress = 0.025) between total microbial community composition of the rhizosphere ($n = 6$) and bulk ($n = 6$) compartments (A) or the low $P_i$ fertilizer (Low P) and high $P_i$ fertilizer (High P) (B) treatments. Each point represents one metagenomic sample ($n = 12$). Data representing relative variable importance (R) and significance (p) calculated by PERMANOVA (ANOSIM) are displayed. (C) Multivariate analysis of the active microbial communities collected from the same soil samples, bulk soil (circles) and rhizosphere soil (squares). (D) The relative abundance of detected proteins in all samples based on label-free quantification (LFQ) values. Pale blue equals the least abundant, black equals the mean abundance, and red equals the most abundant. Dendrograms for both sample and protein were calculated.

Table S2d), suggesting multiple strains of *Pseudomonas* were highly active in the rhizosphere, confirming observations when analyzing the initial DIAMOND searches (Table S2c).

Comparison of individual protein abundance with corresponding ORF abundance in the MG demonstrated plant-associated bacteria, such as *Gammaproteobacteria* (*Pseudomonadaceae*), *Betaproteobacteria* (*Burkholderia*, *Oxalobacteraeae*, *Commonamondaeae*) and *Bacteroidetes* (*Flavobacteriaeae*), were more active in the rhizosphere despite minor changes in gene relative abundance (Fig. 3B). In addition, proteins expressed by methylotrophic bacteria were also enriched in the rhizosphere relative to surrounding bulk soil. For *Acidobacteria* and *Actinobacteria*, while some proteins were more abundant in the rhizosphere, other proteins were more abundant in the bulk soil. This is consistent with the life history strategy of many taxa related to these two phyla (34). Proteins related to various genera associated with the Candidate Radiation Phyla were also detected

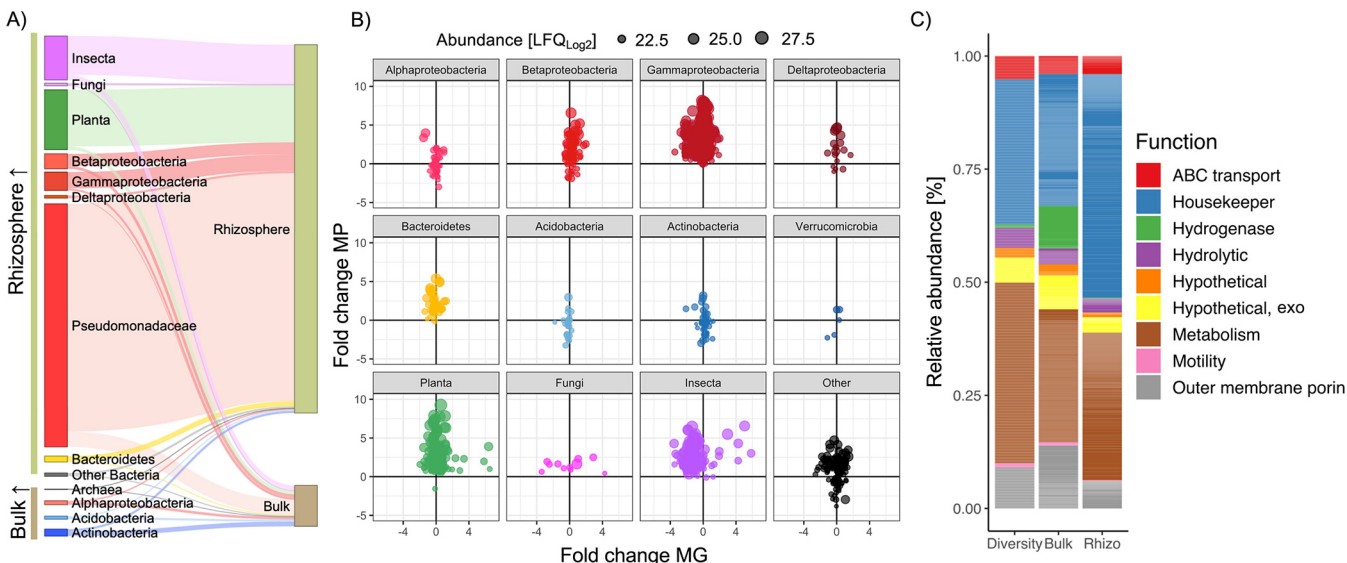

**FIG 3** Taxonomic profile of the *in situ* MEP sampled from field-grown *Brassica napus*. (A) Compartmental partitioning based on the relative abundance of proteins associated with various taxonomic groups identified in either rhizosphere or bulk soil samples. (B) Comparison of relative protein expression versus relative gene abundance in rhizosphere versus bulk soil samples. Each point represents a single protein and its size represents its relative abundance in the MEP. Proteins were partitioned into various taxonomic groups. (C) Broad functional assessment of the bacterial MEP. The relative abundance of these functions was calculated by either counting the total number of distinct detected proteins associated with each function (Diversity) or by determining their relative abundance (LFQ values) in either the bulk (Bulk) or rhizosphere (Rhizo) MEP. Results plotted are the mean of 6 replicates: 3 $P_i$ replete and 3 $P_i$ deplete for each compartment.

demonstrating this large group of enigmatic bacteria are active in natural soils. Together, this demonstrates MEP can identify changes in metabolic activity that can be masked by solely relying on metagenomic data.

Based on functional annotation of all identified bacterial proteins, a large proportion of intracellular proteins were still captured during our extraction step, many of which were related to either central or auxiliary metabolism as well as housekeeping functions and protein synthesis (Metabolism, Housekeeper, Fig. 3C). Despite the greater amount of total protein detected in rhizosphere samples (Fig. 2D), ribosomal proteins (protein synthesis marker) represented a greater proportion of total protein in this compartment, further demonstrating elevated microbial activity in this rhizosphere compartment relative to bulk soil. The most abundant extracellular proteins were related to outer membrane porins, substrate binding proteins associated with ABC transporters and extracellular hydrolytic enzymes. In bulk soil, but not in the rhizosphere, a significant proportion of protein was assigned to hydrogenases.

**Comparison of metagenomic and metataxonomic field-grown *Brassica napus* L. soil communities demonstrates the importance of metaproteomics.** Microbial community composition did not differ significantly between compartment (Fig. 2A), and many rhizosphere-enriched proteins were encoded from genes whose abundance in the MG showed little variation between either soil compartment (Fig. 3B). Therefore, to better determine the relative abundance of active taxa identified in the metaproteome (MP), we expanded our taxonomic assessment of the total soil microbial communities (MG) associated with either bulk soil or rhizosphere soil compartments. To do this, we analyzed the assembled MG using the read abundance and taxonomy of **S**ingle copy **C**ore **G**enes (SCGs), as well as generating an amplicon-based 16S rRNA gene profile (Fig. 4). Both the SCG and 16S rRNA gene profiles showed *Actinobacteria* (MG = 24%; 16S bulk = 48%, and 16S rhizo = 43%) and *Proteobacteria* (MG = 44%; 16S bulk = 21%, 16S rhizo = 29%) numerically dominated the total genomic content of the total soil bacterial community, while *Bacteroidetes* constituted only 5%. At the order level, based on SCGs, separation by compartment (bulk v rhizosphere) did not significantly affect the soil communities' taxonomic structure (Fig. S4). 16S rRNA gene profiles revealed the abundance of *Pseudomonadaceae* (3.43%; 100% *Pseudomonas*) was

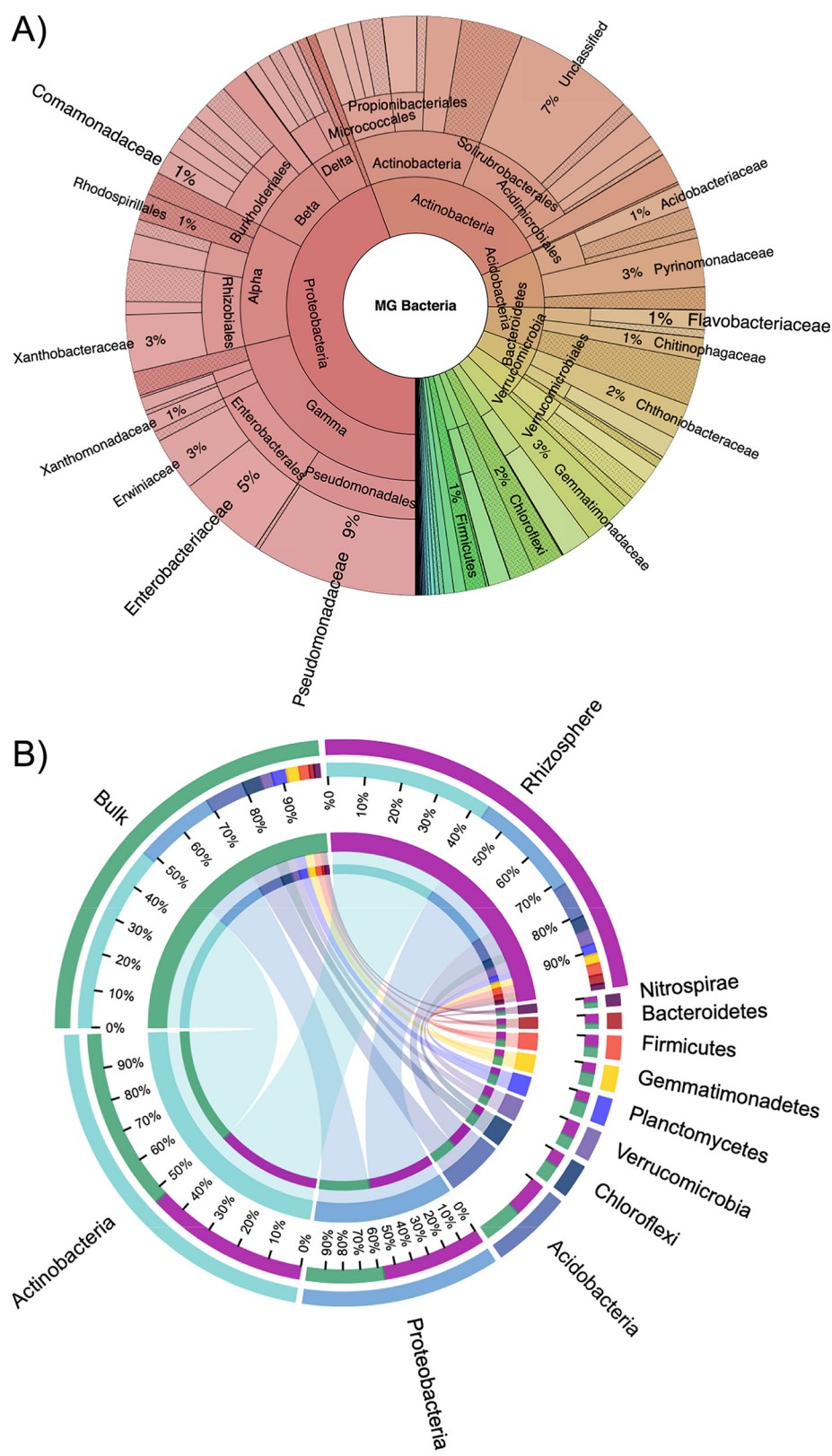

**FIG 4** Composition of bulk soil and rhizosphere soil microbial communities sampled from field-grown oilseed rape based on the composition of single copy core genes in the metagenome or 16S rRNA gene amplicon profiling. (A) Relative abundance of all bacterial taxa in the combined (bulk and rhizosphere) soil metagenome. Selected taxonomic groups of interest in this study are labeled while others have been omitted for clarity. (B) CIRCOS plots showing the relative abundance distribution among the dominant phyla in either the bulk soil or rhizosphere compartment.

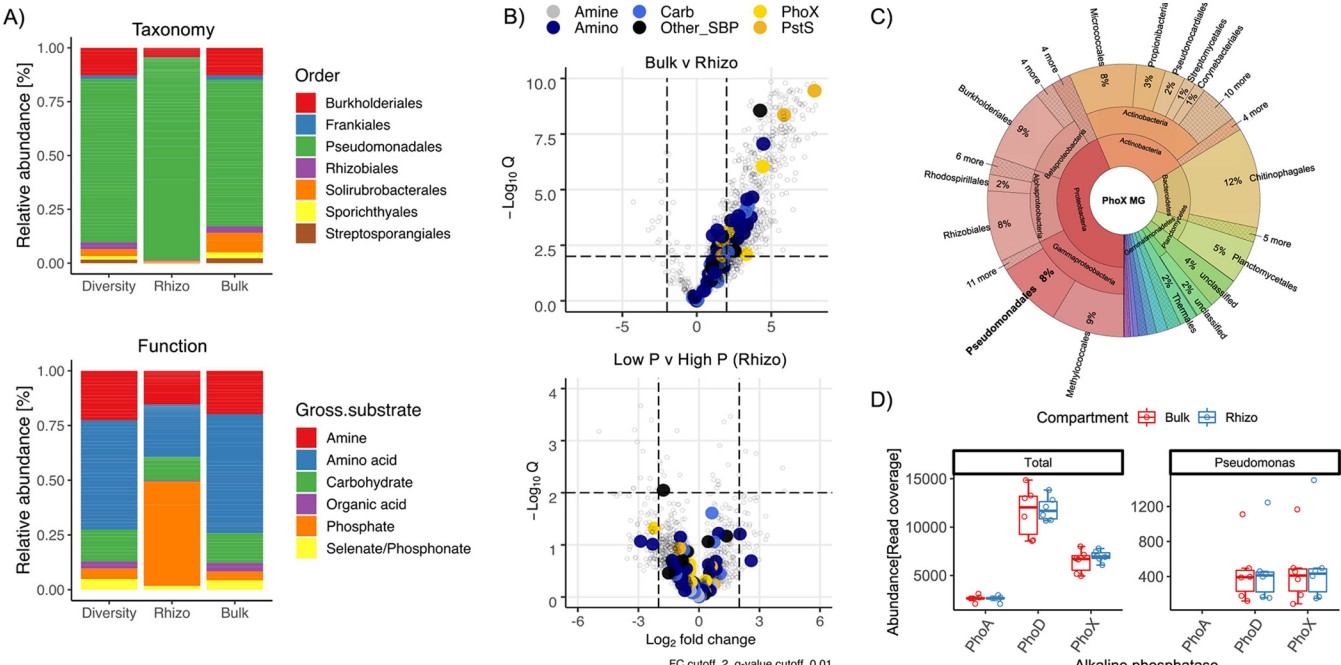

**FIG 5** Functional analysis of *in situ* plant-microbe interactions based on soil metaproteomes. (A) Taxonomy and gross functional classification of substrate binding proteins identified in bulk and rhizosphere (rhizo) soil metaproteomes. The relative abundance of each function was calculated by either counting the total number of distinct detected proteins associated with each function (Diversity) or by determining their relative abundance (LFQ values) in either the bulk (Bulk) or rhizosphere (Rhizo) MEP. (B) The effect of compartment and phosphate fertilizer regime on the expression of bacterial substrate binding proteins. (C and D) Taxonomy (C) and relative abundance (D) of all PhoX ORFs identified in bulk and rhizosphere soil MGs combined.

not significantly different between bulk and rhizosphere compartments, despite significantly greater activity in the rhizosphere, as was the case for SCG-derived abundance data (Fig. 3A). In contrast, rhizosphere-active *Flavobacteraceae* (100% *Flavobacterium*) and *Betaproteobacteria*, which made up 3.8% and 6.12% of the 16S rRNA gene rhizosphere community, respectively, were significantly enriched in this compartment compared to the bulk soil (Fig. S5). In summary, two methods for taxonomic assignment and relative abundance determination demonstrated significantly less shifts between soil compartments based on DNA in comparison with protein.

**Active autochthonous *Pseudomonas* spp. experience $P_i$ limitation under field conditions independently of fertilizer regime.** To better determine the metabolic interactions occurring in the rhizosphere, we focused our analysis on the proportion of proteins predicted to be either periplasmic, outer membrane associated, or extracellular, ignoring predicted cytoplasmic proteins whether they had a role in nutrient acquisition or not (Table S2e). The expression of ABC-transporter related substrate binding proteins is an excellent proxy for metabolic interactions operating in any given environmental niche. The majority of these were expressed by *Pseudomonas* spp. and to a lesser extent *Burkholderiales* spp. (Fig. 5A). In agreement with our laboratory pot experiment inoculated with *P. putida* BIRD-1, many detected substrate binding proteins were associated with predicted amino acid transporters, but a significant number were also predicted to transport other nitrogenous compounds such as polyamines and quaternary amines, as well as carbohydrates (Fig. 5A, bottom). Importantly, in the rhizosphere compartment, two out of three *Pseudomonas* proteins identified as PstS, the substrate binding protein associated with the high-affinity phosphate ABC transporter, were highly expressed (Fig. 5A and B). Indeed, one identified PstS was among the top 10 most abundant proteins in the total MP, including abundant plant and insect proteins (Table S2c). While numerous substrate binding proteins were significantly more abundant in the rhizosphere, fertilizer regime had no significant effect on their expression, even for PstS (Fig. 5A and Table S2c).

In addition, we also detected numerous extracellular hydrolytic enzymes (Fig. 5 and

Table S2c). Among these were five alkaline phosphatases belonging to the PhoX family, all of which were most closely related to the *P. fluorescens* group (Fig. S6). Despite the limited diversity of alkaline phosphatases in the metaexoproteome, *Pseudomonas* PhoX ORFs constituted 2% and 8% of the diversity and richness (relative abundance) in the MG, respectively (Fig. 5C). Furthermore, despite their absence in the metaexoproteome, there were also a significant number of ORFs encoding either the PhoD or PhoA alkaline phosphatases in the MG, with PhoD the most abundant of all three (Fig. 5D). ORFs encoding PhoA and PhoD from a diverse range of taxa were present in both compartments, (Fig. S7) despite no evidence of expression. Finally, the abundance of *Pseudomonas phoD* was almost identical to *Pseudomonas phoX* (Fig. 5D), further suggesting preferential expression of the latter alkaline phosphatase.

## DISCUSSION

Building a holistic understanding of plant-microbe interactions relies on the development of suitable tools to investigate complex and simultaneously occurring metabolic processes *in situ*. Here, first using *P. putida* BIRD-1 as a model and then analyzing natural field-soil microbial communities, we demonstrate that metaexoproteomic assessment of the rhizosphere is achievable and can significantly refine our understanding of the establishment and function of the plant microbiome. Specifically, these data generate further testable hypotheses surrounding the genomic basis of rhizosphere competence and microbial nutrient cycling, which will ultimately guide our ability to engineer plant microbiomes and better determine abiotic and biotic causes of plant disease. Likewise, this method successfully captured extracellular plant and corresponding pathogenic aphid proteins demonstrating the efficacy of this method to also understand plant host-pathogen interactions.

Plants differ in their ability to manipulate soil communities. For example, while a strong "rhizosphere effect," i.e., enrichment of rhizosphere-specific bacteria recruited from the surrounding bulk soil, can be observed for barley (16), other plants elicit much more subtle differences (15). A significant limitation of metagenomics is its inability to clearly identify the most active microbes and metabolic processes occurring in a specific environment, which is further compromised with the inclusion of subtle spatiotemporal parameters. In addition, while metatranscriptomics can provide information on activity, it suffers from neglecting posttranscriptional and posttranslational regulation of protein synthesis, as well as an inability to spatially resolve protein compartmentalization, i.e., intra- versus extracellular location. While our metagenomic data were consistent with the observation that oilseed rape elicits a weak rhizosphere effect on soil microbial communities (14), by utilizing MEP we observed a clear difference between the active microbial community present in the rhizosphere compared to the surrounding bulk soil. This agrees with metatranscriptomics studies investigating the active plant microbiota in various crops (35). This difference in activity can be largely attributed to an increase in the quantity of microbial and plant protein captured in the rhizosphere and elegantly demonstrates the rhizosphere as a hot spot for plant-induced microbial activity (1, 2, 36–38).

Partitioning the MEP between compartments was not only achieved through the capture of significantly more bacterial protein in the rhizosphere (Fig. 2B) but also a shift in the taxonomic groups producing these proteins. Rhizosphere-specialized bacterial taxa can also be thought of as copiotrophs, responding to elevated labile and complex organic carbon deposition. On the other hand, oligotrophs are relatively more active in carbon-depleted bulk soil (34). The large increase in *Pseudomonas* activity as well other bacterial groups such as *Flavobacterium* (*Bacteroidetes*), and various *Burkholderiales* (*Betaproteobacteria*) in the rhizosphere is consistent with their predicted life history strategies (34, 39). Likewise, our MEP data also confirmed that "bulk soil-specialized" or oligotrophic bacteria, such as those related to *Verrucomicrobia*, *Actinobacteria*, and *Acidobacteria* (39), were relatively more active in the surrounding soil. While our study only captured a single time point, our data clearly revealed that various distinct strains of *Pseudomonas* are highly active in the rhizosphere of oilseed rape and represent a major and ecologically

important component of this crop's microbiome (14, 40, 41). *Pseudomonas* represents a relatively small fraction of the seed microbiome. Thus, an increase in their relative abundance, especially several strains, in both the rhizosphere and root during the early stage of oilseed rape growth indicates active selection from the surrounding bulk soil (40, 41). Further investigation should now focus on determining how the active fraction of the community changes over time, particularly at different growth stages. This would help determine the stability of field-grown plant microbiomes, which could ultimately be used to determine early signs of pathogen-induced dysbiosis.

In addition to gaining functional insight into the plant microbiome through identification of active taxonomic groups, we also identified numerous proteins related to various beneficial functions (1, 36). Plant root exudates shape microbial communities and amino acids can become the major group of exudates released by plants over time and, in addition to quaternary amines, can represent a significant fraction of the dissolved organic N pool (42–44). Furthermore, the turnover of the soluble amino acid pool in soil may be orders of magnitude greater than that of ammonium or nitrate (45). Based on our MP data collected from both our inoculated pot experiment and field trial, we discovered *Pseudomonas* metabolism shifts toward the turnover of amino acids and other N-containing compounds when growing in the oilseed rape rhizosphere. Given that most heterotrophic soil microbes are carbon limited, the high expression of uptake and catabolic proteins targeting amino acids and other nitrogenous carbon sources (predominantly amines) observed here suggests that microbial-mediated mineralization of ammonium may be a key process in the rhizosphere, as observed in marine systems (31, 46). Thus, release of nitrogenous organic carbon exudates may represent a mechanism that allows plants to get an immediate return on their metabolic investment in the form of labile ammonium. This aligns with the idea that plants "prime soils" for microbial N mineralization through the exudation of organic C, stimulating expression of peptidases and proteases (45).

Plant-available phosphate is often a small fraction of the total soil P content. The slow diffusion of $P_i$ in soil means that plant uptake during growth creates a zone of $P_i$ depletion around the roots (1 to 3 mm) (47–50), which is only intensified by increases in microbial growth on plant-derived labile organic carbon (1). In almost all bacteria, including *Pseudomonas*, synthesis of phosphatases and the high affinity phosphate transporter PstSABC is negatively regulated by exogenous levels of inorganic orthophosphate. Thus, these proteins serve as excellent markers to assay for phosphate depletion (51–53). Furthermore, elevated soil phosphatase activity has recently been shown to co-occur with the severity of $P_i$ depletion (54). While our pot experiments showed no evidence of localized $P_i$ depletion in the rhizosphere, in our field experiment the identification of five and three distinct *Pseudomonas* PhoX and PstS homologs, respectively, suggests rhizosphere-dwelling *Pseudomonas* spp. experience phosphate-limiting growth conditions, despite saturation of the soil with inorganic fertilizers. PhoD is commonly used as the major gene marker for microbial phosphatase activity (55–57). However, despite this family being the most abundant phosphatase in the MG for both the total community and *Pseudomonas* population, only PhoX was detected in the MEP, consistent with its role as the major phosphatase in plant-associated *Pseudomonas* (28, 53, 58) and other environmental *Proteobacteria* (59, 60). Furthermore, stimulation of *Flavobacteriia* likely has importance consequences for remineralization of organic P (27, 61).

**Conclusions.** Here, we present the first metaexoproteomic assessment of the plant microbiome sampled from a field-grown agricultural crop. Our new technique enabled us to identify highly active taxa in the rhizosphere and the key nutrients they target. Crop production heavily relies on the unsustainable use of inorganic N and P fertilizers, and modern agricultural initiatives are moving toward the use of more sustainable organic sources of either N or P. The success of this strategy is dependent on having a deep understanding of the key microbial players involved in N and P cycling and the biotic and abiotic factors that control this. In this regard MEP can greatly advance our understanding of the

spatiotemporal dynamics of functionally important taxa and allow us to better engineer the plant microbiome through environmental and plant genotypic selection.

## MATERIALS AND METHODS

**Growth conditions for laboratory pot experiments.** Plants were grown in a field soil collected from the University of Reading's Sonning Farm facility (51° 28′ 55.3836″ N, 0° 53′ 44.3688″ W) between a depth of 20 and 60 cm. The soil properties are 50% sand, 38% silt and 12% clay, with a pH of 6.7. Soil was air dried for 72 h, sieved through a 1-cm sieve, and supplemented with 0.4 g L$^{-1}$ NH$_4$NO$_3$, 0.75 g L$^{-1}$ KNO$_3$, and 0.225 g L$^{-1}$ Ca(H$_2$PO$_4$)$_2$. Pots were filled with 1 L of soil, autoclaved, and left for 5 days. A 1-cm layer of moist perlite was overlaid for the seeds to germinate in. Two seeds of *Brassica rapa* R-o-18 were sown into the perlite layer to germinate, and thinned to one plant per pot. At sowing, pots were treated with 100 mL minimal media with *Pseudomonas putida* BIRD-1. *P. putida* BIRD-1 was maintained on Luria Bertani (LB) agar (1.5% wt/vol) medium at 30°C. Prior to inoculating the pots, BIRD-1 was grown overnight in LB at 30°C and used to inoculate (1% vol/vol) minimal medium using 20 mM glucose as the sole carbon source, 7.5 mM ammonium as the sole nitrogen source, and 1 mM phosphate as the sole P source (27). Cells were incubated at 30°C (shaking at 160 rpm) and grown to a final yield of 10$^9$ cells mL$^{-1}$.

Pots were placed in large seed trays and watered from below using deionized water throughout the experiment. Pots were placed in a controlled environment growth room (Units 37–38; Weiss Technik UK Ltd., Loughborough, UK), with a 16-h day length, 21°C day, 18°C night, and 80% relative humidity. Light was provided by a bank of fluorescent bulbs with a photosynthetic photon flux density of 250 $\mu$mol m$^{-2}$ s$^{-1}$. After 4 weeks growth, plants were harvested. Plants were gently extracted from the soil and excess soil shaken off the root system. Roots were cut and placed in a 50-mL falcon tube containing 20 mL potassium sulfate buffer (PSB; 0.5 M K$_2$S$_2$O$_4$, 10 mM EDTA, pH 6.6). The roots were vortexed in the PSB for 10 s to remove rhizosphere soil before being transferred to a fresh tube containing PSB and vortexed again for 10 s. The rhizosphere soil from both tubes was combined into one sample, centrifuged at 4,000 rpm at 4°C for 5 min and then snap-frozen in liquid nitrogen. Samples were stored at −80°C prior to being freeze-dried.

**Field sampling site and conditions.** *Brassica napus* L. plants were sampled at the four-leaf growth stage in October 2017, using a systematic sampling design (Fig. S1), from the same location as the soil was collected for the pot trials above. Plants were removed from the soil, and the roots were processed as described in the previous section.

**Extraction of extracellular proteins from soil.** To extract extracellular proteins from agricultural field soil, the methods developed by Rollings-Johnson et al. (29) and Armengaud et al. (26) were modified to account for the reduction in available sample associated with rhizosphere soil. Briefly, loose soil was shaken off plant roots and discarded, and the remaining rhizosphere soil was removed from the roots by immersion and shaking in a 0.5 M KSO$_4$ 10 mM EDTA buffer, pH 6.6, until approximately 30 g of soil had been collected in a 1:3 wt/vol ratio of soil: buffer. This solution was incubated at room temperature with 100 rpm shaking for 1 h, centrifuged at 12,800 × *g* for 20 min at 4°C, decanted into Nalgene centrifuge tubes, and centrifuged at 75,600 × *g* for 20 min at 4°C. The supernatant was then sequentially filtered through 0.45- and 0.22-$\mu$m pore-size PVDF filters (Fisher Scientific) to remove any bacterial cells and adjusted to pH 5 with 10% vol/vol trifluoroacetic acid. Then, 0.001% (vol/vol) of StrataClean resin (Agilent Technologies, UK) was added in order to bind proteins, and samples were incubated in a rotatory shaker at 4°C overnight. Samples were centrifuged at 972 × *g* for 5 min at 4°C, and supernatants were discarded. If any precipitates were observed, then the resin was resuspended in dH$_2$O adjusted to pH 5 with 10% vol/vol Trifluoroacetic acid, and this centrifuge step was repeated. Next, the resin was resuspended in 20 $\mu$L of 1× lithium dodecyl sulfate 1× dithiothreitol gel loading buffer (Expedeon-Abcam, UK), heated to 95°C for 5 min, and then sonicated in a water bath for 5 min, twice in succession.

**Identification and quantification of proteins.** For protein identification a short run (~2 min) was performed to create a single gel band containing the entire exoproteome, as previously described by Christie-Oleza et al. (62). In-gel reduction was performed prior to trypsin digestion and subsequent clean up as previously described (62). Samples were analyzed by means of nanoLC-ESI-MS/MS using an Ultimate 3000 LC system (Dionex-LC Packings) coupled to an Orbitrap Fusion mass spectrometer (Thermo Scientific, USA) using a 60-min LC separation on a 25-cm column and settings as previously described (63).

To identify peptides, we used an iterative database search approach. First, all detected mass spectra were searched against the total assembled metagenome (MG) database containing 64.1 M open reading frames (ORFs), generated from a composite metagenome of the field soil, detailed below. To reduce redundancy, ORFs were clustered at 90% using CD-HIT and representative ORF sequences were used as the database. A 90% clustering value was chosen based on a preliminary database search using a subset of the total MG (0.931 M ORFs), focusing on P cycling and other extracellular proteins (Pset). This reduced database was clustered at both 99% and 90%, and while some resolution was lost at 90% clustering, the vast majority of protein clusters were identified. Therefore, 90% clustering was applied to constrain the total MG database. X!-Tandem and MS-GF+ searches were performed, generating a database of 206,065 identified proteins, prior to FDR and minimum unique peptide filtering. This reduced ORF database was then used in a MaxQuant search, returning 6,718 proteins (plus 71 decoy and 21 contaminants). Removal of proteins with only one observed peptide, only identified by modified peptides, and allowing for a peptide threshold FDR of 5% and a protein threshold FDR of 10% resulted in a final protein detection of 1,895 protein groups (10 contaminants). Setting FDR thresholds is a complex issue when applying these to metaproteomics (64). After manual scrutiny (visualized using Scaffold Viewer

v4.8.7) of search results obtained using the Pset database and setting the FDR at either 5% (Table S2a) or 10% (Table S2b), we decided on using a relaxed FDR of 10%. The extra peptides/proteins identified by relaxing the FDR threshold did not occur in a random fashion, suggesting that true biological inference was maintained in the vast majority of cases. Indeed, most extra identified peptides were assigned to proteins already identified using a more stringent FDR, suggesting that isoforms and microdiversity of proteins at the amino acid level restricted the number of peptides/proteins identified. This is perhaps a consequence of our clustering step, but it should be noted that the overral biological inference remains remarkably similar for either chosen FDR threshold (Table S2a and b). The highest ranked protein in each group, based on number of unique peptides and/or probability, was taken forward. Typically, protein groups consisted of proteins of identical function separated by taxa, predominantly at the species level. *In silico* protein sorting prediction tools, such as secretomeP 2.0, LipoP, and SignalP (https://services.healthtech.dtu.dk/), were used to determine if proteins were truly extracellular or simply intracellular and released during cell lysis/death. Quantification, statistical analyses, and data visualization of exoproteomes were carried out in Perseus and RStudio (version 1.2.5033). The mass spectrometry proteomics data have been deposited in the ProteomeXchange Consortium via the PRoteomics IDEntifications (PRIDE) partner repository with the data set identifiers accessions PXD033692 (field soil MEP, using the Pset database) and PXD033802 (pot soil MEP).

**Extraction of metataxonomic and metagenomic data.** DNA from either bulk or rhizosphere soil was extracted using the FastDNA Spin Kit (MP Biomedicals) soil extraction kit following the manufacturer's instructions. All samples were checked for integrity and quality by gel electrophoresis (1% wt/vol agarose) and NanoDrop Spectrophotometry (ThermoFisher). DNA was quantified using QuBit (ThermoFisher). For 16S rRNA gene profiling of the microbial communitiesm, 16S rDNA amplicons covering the V1-3 variable regions were amplified using 27F and 534R eubacterial primers with Illumina overhang adapter sequences. Following PCR cleanup (as per the manufacturer's instructions) using AMPure XP beads (Beckmann Coulter), indices were attached using the Nextera XT index kit (Illumina) as per the manufacturer's instructions. Amplicons were quantified, pooled, and prepared for 2x300bp paired end sequencing using an Illumina Miseq platform, as per the manufacturer's instructions. For shotgun-metagenomes, libraries and sequencing were performed by Novagene Ltd using an Illumina HiSeq–PE 150 bp.

Metataxonomic assessment of microbial communities using the 16S rRNA marker gene was performed using QIIME2 (version 2020.11) (65). Singleend (forward reads) files were demultiplexed using the demux plugin. Then, quality control and denoising were performed using the Dada2 plugin (66). All amplicon sequence variants (ASVs) were aligned with mafft and used to construct a phylogeny with fasttree2 (via q2-phylogeny) (67). Taxonomy was assigned to ASVs using the q2-feature classifier against the Greengenes 13_8 97% reference sequences (68). Raw sequence data have been deposited in the NCBI Sequence Read Archive (SRA) under bioproject PRJNA738866.

A Bray-Curtis dissimilarity matrix was calculated based on each samples clustered-ASV profiles and used for nonmetrical multidimensional (NMDS) scales. We have modeled the distances between UniFrac and the Bray-Curtis discrepancies using the ASV-level table through one-way similitude analysis (ANOSIM) to investigate differences in community composition between rhizosphere and soil compartments. All of the above models were constructed using RStudio (version 3.6), including analyses for comparing the relative abundance of different bacterial taxonomic levels.

**Coassembly of the composite metagenome.** Three different coassembly iterations were explored, either grouping samples by fertilizer treatment, biological compartment (bulk versus rhizosphere), as well as full coassemblies. Statistical analyses, such as N50 and total assembly size, determined the full coassembly providing the highest quality assembly of metagenomics reads. All assemblies were carried out using megahit (69) (version 1.1.3). After removing contigs smaller than 500 nucleotides, ORFs were called using prodigal (version 2.6.2) (70) with option "-p meta." This resulted in a collection of 64.1 million different ORFs. ORFs multisample coverage profiles were generated by mapping all sample reads to the assembly using both bwa-mem (version 0.7.17-r1188) (71) and samtools (version 1.10) (72).

**Metagenomic taxonomic profiling.** Assembly of the 16S rRNA gene from shotgun metagenomic read data is notoriously poor and also suffers from susceptibility to highly variable copy number across bacterial taxa. Therefore, community composition of the MG was based on the taxonomy and abundance of **S**ingle copy **C**ore **G**enes (SCG). For this, the database of 64 millions ORFs was annotated using rpsblast (version 2.9.0+) (73) using the pssm formatted COG database (74), which is made available by the CDD (75). For the set of 36 COGs taken as SCGs, corresponding ORFs were clustered at 5% ANI using mmseqs2 (version 13.45111) (76) with options "easy-cluster," "-cov-mode 2," "-max-seqs 1000," and "-c 0.80." This resulted in a median number of 6,280 clusters over the 36 SCGs. After adding sequences from Refseq genome representatives, a series of 36 corresponding phylogenetics trees were built using in sequence, mafft (version v7.407) (77), trimal (version v1.4.rev22) (78) with options "-gt 0.9" and "-cons 60" and FastTree version 2.1.10 (67). Using the python library ete3 (version 3.1.2), each SCG cluster was assigned to the nearest refseq representative. Taxonomic profiles were obtained by summing the SCG cluster ORFs coverage along taxonomic assignment. In the case where more than one of the 36 SCGs was found at the same taxonomic level, median coverage was taken. Normalization was carried out by taking for each sample the median total coverage of the 36 SCGs. This approach is insensitive to the varying size of genomes, so that organisms with larger genomes do not appear to be more abundant than those with smaller genomes.

To identify individual phosphatases (PhoX, PhoD, and PhoA) the methods developed in reference 79 were applied. Briefly, this involved performing a hmmsearch of the assembled MG ORF database, using pregenerated profile hidden Markov models (pHMMs) for each protein (67). To assign taxonomy, identified ORFs were then aligned by BLASTP (e$^{-20}$) to a manually curated database generated from all sequences

deposited in the IMG/JGI database. Any sequences (<2%) not aligning to any predicted phosphatases were removed from the data set. The read coverage for each ORF was then extracted to determine the relative abundance of each ORF. ORF coverage was then normalized by accounting for variation in total SCG coverage as per Lidbury et al. (79). For PhoX, phylogenetic analyses were performed using IQ-Tree using the parameters -m TEST -bb 1000 -alrt 1000. Representative PhoX sequences obtained from the genomes of isolates spanning the diversity of *Pseudomonas* were aligned with the sequences identified in the MEP. Evolutionary distances were inferred using maximum-likelihood analysis. Relationships were visualized using the online platform the Interactive Tree of Life viewer (https://itol.embl.de/). Raw sequence data have been deposited in the NCBI SRA under bioproject PRJNA738866.

## SUPPLEMENTAL MATERIAL

Supplemental material is available online only.

**FIG S1**, EPS file, 0.8 MB.
**FIG S2**, EPS file, 0.1 MB.
**FIG S3**, EPS file, 0.4 MB.
**FIG S4**, EPS file, 0.6 MB.
**FIG S5**, EPS file, 0.2 MB.
**FIG S6**, EPS file, 0.1 MB.
**FIG S7**, PDF file, 0.2 MB.
**TABLE S1**, XLSX file, 0.1 MB.
**TABLE S2**, XLSX file, 1.3 MB.

## ACKNOWLEDGMENTS

We thank the Warwick Proteomics Research Facility, namely Dr. Cleidiane Zampronio for assistance in generating and processing the mass-spectrometry data.

This study was funded by the Biotechnology and Biological Sciences Research Council (BBSRC) and National Environmental Research Council (NERC) under project codes BB/L026074/1, BB/T009152/1 and NE/S013539/1 linked to The Soil and Rhizosphere Interactions for Sustainable Agri-ecosystems (SARISA) program and a Discovery Fellowship (to I.D.E.A.L.) and NERC Environmental 'Omics Synthesis Grant (to I.D.E.A.L., J.P., and E.M.H.W.), respectively.

We declare no competing interests.

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
