## [Reviewer comments · mSystems]

Stimulation of distinct rhizosphere bacteria drive phosphorus and nitrogen mineralisation in oilseed rape under field conditions

Ian Lidbury, Sebastien Raguideau, Chiara Borsetto, Andrew Murphy, Andrew Bottrill, Senlin Liu, Richard Stark, Tandra Fraser, Andrew Goodall, Alex Jones, Gary Bending, Mark Tibbett, John Hammond, Chris Quince, Dave Scanlan, Jagroop Pandhal, and Elizabeth Wellington

Corresponding Author(s): Ian Lidbury, The University of Sheffield

Review Timeline:

Submission Date:	January 12, 2022
Editorial Decision:	March 5, 2022
Revision Received:	May 23, 2022
Accepted:	June 12, 2022

Editor: Michelle Heck

Reviewer(s): Disclosure of reviewer identity is with reference to reviewer comments included in decision letter(s). The following individuals involved in review of your submission have agreed to reveal their identity: Jean Armengaud (Reviewer #2)

Transaction Report:

DOI: <https://doi.org/10.1128/msystems.00025-22>

March 5, 2022

Dr. Ian Dennis Edmund Alan Lidbury
The University of Sheffield
Department of Animal and Plant Science
Sheffield
United Kingdom

Re: mSystems00025-22 (Metaexoproteomics identifies active plant-microbe interactions operating in the rhizosphere)

Dear Dr. Ian Dennis Edmund Alan Lidbury:

Thank you for submitting your manuscript to mSystems. We have completed our review and I am pleased to inform you that, in principle, we expect to accept it for publication in mSystems. However, acceptance will not be final until you have adequately addressed the reviewer comments.

This paper has more than 10 supplementary files. That number has to be reduced to less than 10. I suggest merging some of the supplementary tables and/or merging figures S2 and S3 into a single figure.

Please add a section entitled "Data Availability" at the end of the Materials and Methods section and move the sentences providing information about data availability into this section.

Preparing Revision Guidelines

Sincerely,

Michelle Heck

Editor, mSystems

Journals Department
Reviewer comments:

Reviewer #2 (Comments for the Author):

The manuscript proposed by Lidbury et al. identifies active plant-microbe interactions in the Brassica rhizosphere by means of metaexoproteomics. While analyzing exoproteomes in soils is a difficult task, the authors have convincingly demonstrated how this methodology is far more informative than metagenomics. This is impressive work that should be a motivation for others interested in plant & microorganisms interactions.

Here are some points to consider to further improve the manuscript:

1. The experiment regarding the extraction method may suffer from bias. Spiking protein standards in soil samples may not be exactly the same as proteins trapped for some time in soil or in specific biofilm-like structures. This bias may be taxon-specific. The authors could briefly discuss this point.
2. The identified *Pseudomonas* proteins designated *P. fluorescens* as the main player. I wonder if a peptide-centric analysis (LCA approach) shows the same prominent species. In addition, it is not indicated whether the 16S rRNA barcode results confirm this trend.
3. Metaproteomics interpretation is based on a metagenome-derived database. The methodology is sound, but I wonder if a more refined interpretation could improve the depth of analysis, such as removing low number sequence reads and giving more importance to the most abundant microorganisms in the database prior metaexoproteomics interpretation.
4. Only bacterial proteins are listed and commented on, while probably unicellular eukaryotes are also present in the samples. Is it a bias due to the database or the interpretation pipeline? Is it an important item that requires some words in the discussion?
5. The PRIDE identifier is not provided in the manuscript and I could not check the quality of the dataset. As the authors are experienced in metaproteomics, the quality of the dataset should not be a problem at all.
6. Is there any effect of low temperature during centrifugation for 5 min at 4{degree sign}C on the proteome of the rhizosphere soil (i.e. cold shock induced proteins)?
7. The authors mentioned twice « in its infancy »... (importance & introduction sections). Although this formulation is widely used by scientists, I am not sure such expression is appropriate in these specific sections. Such wording could apply to almost all scientific field because, logically, time allows significant advances and gains in maturity. However, its use could mean that previous observations were not so valid, diminishing the value of earlier pioneering work.
8. Line 382: a percentage is given with 4 significant digits. Please, consider the variability of your measurement (how many 16S rRNA measurements were performed per sample?).
9. What was the amount of *Pseudomonas putida* BIRD-1 used as inoculum for the overnight culture on minimal medium, temperature and shaking?
10. In the M&M section, abbreviations such as RH, PPF, ... can be difficult for non-specialists.

Reviewer #3 (Comments for the Author):

This manuscript outlines implementation of an experimental/informatics method to examine the capture and identification of extracellular proteins (termed metaexoproteomics or MEP) synthesized in the rhizosphere of Brassica spp. Their MEP results revealed a significant shift in the metabolically active fraction of the soil microbiota responding to the presence of *B. napus* roots that was not apparent in the composition of the total microbial community. The authors contend that this metaproteomic assessment of the 'active' plant microbiome at the field-scale demonstrates the importance of moving past a genomic assessment of the plant microbiome in order to determine ecologically important plant-microbe interactions underpinning plant health.

In general, this study is well-designed, systematic, and relies on state-of-the-art approaches in most cases. As it typical in most omics studies, a large amount of data is generated, much of which is somewhat intractable to mine. Nevertheless, the authors have done a nice job in focusing on the extracellular proteins in these natural systems. The manuscript is clearly written, logical in design, and appears scientifically sound. Some specific comments are listed below:

1. I would recommend that the authors consider revising the title a bit to make it more engaging to the lay reader. The current

version is a bit bland - maybe add something that provides a bit of "science nugget" about what was learned here.

2. Page 2, line 30 - might be good to clarify here if these 1885 proteins are exclusively microbial or also include plant ones?
3. Page 3 - here or elsewhere, it might be important to clarify why focus on "extracellular proteins" rather than the cytosolic ones? Why not examine the entire microbes rather than only their extracellular proteins? Wouldn't this give a more complete glimpse of the entire protein machinery (i.e. functionality) of the system?
4. Page 5, line 91 - it might be worth elaborating here that microbial complexity in soils is usually greater than any other environments and leads to considerable problems in metagenome sequencing and assembly, which are critical for quality metaproteome measurements.
5. Page 6, line 116 - would there be an expectation to see extracellular proteins that would reveal information about microbial "stress" or signaling, in addition to response to elevated carbon? Would this information naturally fall-out of the measurements or would that have to be mined specifically?
6. Page 6, line 131 - the native soil undoubtedly contains a natural microbial community. What is known about this and how might this respond to the media addition and/or the addition of the *P. putida*?
7. Line 158 - how do the authors know that the 1 hour incubation at room temperature (without the plant) does not change the metabolic activities of the microbes still present in the soil/buffer solution and thus possibly confound the effects?
8. Line 181 - would be helpful to give a brief description and defense for ORF clustering at 90% - what resolution is lost.
9. Line 187 - an FDR of 10% seems unacceptably high. This would be unusual, so the authors need to provide more detailed rationale and defense for this choice. I think most readers would find this too liberal, suggesting that many of the IDs may be suspect.
10. Line 245 - how does this taxonomic profiling compare to the attempt to assemble MAGs from metagenome data? Did the authors consider MAG assembly, even though it would be variable across the range of taxa?
11. Line 278 - was the sand:soil mix sterilized to kill endogenous microbes? If not, how would this affect the measurement? Should the searching have been done with a larger "pseudo-metagenome" to provide IDs for other members? What is the possibility of redundant peptides/proteins from other microbes in the endogenous community?
12. Line 334 - how close are these genomes - species, strains?
13. Line 353 - the presence of intracellular proteins, while not surprising, suggested that normal cellular lysis is occurring. How do the authors sort out which proteins are definitively "extracellular" from those that simply result from cell death/lysis?
14. Line 438 - might be worthwhile pointing out here that while meta-transcriptomic studies also provide "functional information about the microbiome," they lack the ability to spatially resolve intracellular vs. extracellular activities.
15. Line 453 - do the authors feel that their systems are relatively stable and thus a single time-point measurement is likely to reflect sustained activity vs. temporal "blips"?
16. Line 741 - the use of NSAF represents an older version of relative quantification. This is not to say that it is incorrect but more recent methods to exploit extracted peak areas are more tractable with high resolution LC separations coupled to high mass accuracy MS systems. These also afford deeper dynamic range. The authors should address their choice of NSAF over peak areas?

Reviewer comments:

Reviewer #2 (Comments for the Author):

The manuscript proposed by Lidbury et al. identifies active plant-microbe interactions in the Brassica rhizosphere by means of metaexoproteomics. While analyzing exoproteomes in soils is a difficult task, the authors have convincingly demonstrated how this methodology is far more informative than metagenomics. This is impressive work that should be a motivation for others interested in plant & microorganisms interactions.

Thank you for your kind words.

Here are some points to consider to further improve the manuscript:

1. The experiment regarding the extraction method may suffer from bias. Spiking protein standards in soil samples may not be exactly the same as proteins trapped for some time in soil or in specific biofilm-like structures. This bias may be taxon-specific. The authors could briefly discuss this point.

This is a valid point. However, the spiking experiment was simply a proof of principle and we do not draw any major conclusions from this. Our line of thought for the whole method development was two initial spiking experiments: BSA and exoproteins captured in vitro. Once this showed promising results, we then performed the inoculant-based experiment where *Pseudomonas putida* was added in upon planting the seedlings. Exoproteomics was then performed after 3-4 weeks incubation. Hence, the data drawn from this is based around proteins produced, secreted, or escaped from cells, and remaining in the soil matrix during that period.

2. The identified *Pseudomonas* proteins designated *P. fluorescens* as the main player. I wonder if a peptide-centric analysis (LCA approach) shows the same prominent species. In addition, it is not indicated whether the 16S rRNA barcode results confirm this trend.

Sorry, I think our explanation was not completely clear and we have re-worded this. This was a restricted search against four 'well-known' genomes. Diamond searches do indeed show that even using our MG-centric approach, many of the proteins have high identity to multiple *Pseudomonas* strains, particularly those isolated from crop rhizospheres. However, using the phylogenetic classification of *Pseudomonads* (see Jones et al., 2021), these strains would still belong to the *fluorescens* group. We did try and reconstruct genomes from the data, but for some unknown reason, we did not recover decent genomes from the contigs.

We have changed the following section:

'The identified *Pseudomonas* proteins were aligned to four different *Pseudomonas* genomes, representing four distinct *Pseudomonas* groups: *P. putida* BIRD-1, *P. fluorescens* SBW25, *P. stutzeri* DSM4166, and *P. syringae* DC3000. On average, detected proteins had the highest identity with *P. fluorescens* SBW25 (93%), suggesting most of the identified proteins belonged to strains within the *P. fluorescens* group. However, there was significant variation in average identity (%) related to each strain and numerous proteins were absent

from each individual genome (Fig. S3, Table S2d), suggesting multiple strains of *Pseudomonas* were highly active in the rhizosphere, confirming observations when analysing the initial DIAMOND searches (Table S2c).'

Reference:

Jones, R.A., Shropshire, H., Zhao, C. et al. Phosphorus stress induces the synthesis of novel glycolipids in *Pseudomonas aeruginosa* that confer protection against a last-resort antibiotic. *ISME J* 15, 3303–3314 (2021).

3. Metaproteomics interpretation is based on a metagenome-derived database. The methodology is sound, but I wonder if a more refined interpretation could improve the depth of analysis, such as removing low number sequence reads and giving more importance to the most abundant microorganisms in the database prior metaexoproteomics interpretation.

This is a good idea and something that we did not explicitly think of. However, this approach may cause some loss of information when analysing the bulk soil data. For example, some of the proteins detected in the bulk come from less abundant bacteria and may have been removed from the database using this approach. We do acknowledge this would be a useful comparison to conduct in future studies and thank the reviewer for this suggestion. We did perform several preliminary database searches using reduced versions of the metagenome by pulling out proteins using hmm searches, targeting different nutrient cycling functions. However, we found that the 2-step iterative approach managed to capture the same proteins and more.

4. Only bacterial proteins are listed and commented on, while probably unicellular eukaryotes are also present in the samples. Is it a bias due to the database or the interpretation pipeline? Is it an important item that requires some words in the discussion?

We do indeed capture both plant and plant pest proteins in this rhizosphere. This is briefly mentioned on line 125:

'In addition to capturing extracellular plant and aphid-pest proteins in the rhizosphere, we observed greater microbial activity.....'

5. The PRIDE identifier is not provided in the manuscript and I could not check the quality of the dataset. As the authors are experienced in metaproteomics, the quality of the dataset should not be a problem at all.

Sorry, we did not have this processed during the initial submission, the data has now been deposited in the PRIDE database under the accessions PXD033692 (field MEP) and PXD033802 (pot MEP).

We have added this information to our methods section:

'The mass spectrometry proteomics data have been deposited in the ProteomeXchange Consortium via the PRoteomics IDentifications (PRIDE) partner repository with the dataset identifiers accessions PXD033692 (field soil MEP) and PXD033802 (pot soil MEP).'

6. Is there any effect of low temperature during centrifugation for 5 min at 4{degree sign}C on the proteome of the rhizosphere soil (i.e. cold shock induced proteins)?

Prior to this step the potassium sulfate buffer was added. During the methods development of this buffer (Rollings-Johnson et al., 2014), it was observed that this buffer was the best at preserving protein samples and maximising extraction yields. It was observed that buffer and chilled centrifugation helps minimise any unwanted expression during these processing steps. This was only performed for the pot experiment and whilst we cannot rule out there may have been some amount of proteomics response, we did not find any evidence based on our analysis of the *P. putida* exoproteome.

7. The authors mentioned twice « in its infancy »... (importance & introduction sections). Although this formulation is widely used by scientists, I am not sure such expression is appropriate in these specific sections. Such wording could apply to almost all scientific field because, logically, time allows significant advances and gains in maturity. However, its use could mean that previous observations were not so valid, diminishing the value of earlier pioneering work.

Thank you for this insightful comment. We have amended the two sentences as follows:

Unlike its application in seawater^{18, 19}, anaerobic digestors^{20, 21} or the human or animal gut^{22, 23}, soil metaproteomics has been relatively underexploited²⁴. This is partly due to conventional soil extractions methods co-extracting contaminating substances, such as organic carbon and humic acids²⁴.

8. Line 382: a percentage is given with 4 significant digits. Please, consider the variability of your measurement (how many 16S rRNA measurements were performed per sample?).

We thank the reviewer for this. We agree this is excessive. We have shortened all % to 3. We had six biological replicates for each compartment (rhizosphere v bulk), made up of three high Pi and three low Pi treatments. As stated in the methods each individual sample was the conglomerate of at least three plants.

9. What was the amount of *Pseudomonas putida* BIRD-1 used as inoculum for the overnight culture on minimal medium, temperature and shaking?

We have amended the methods as follows:

'Prior to inoculating the pots, BIRD-1 was grown overnight in LB at 30°C and used to inoculate (1% v/v) minimal medium using 20 mM glucose as the sole carbon source, 7.5 mM ammonium as the sole nitrogen source, and 1 mM phosphate as the sole P source²⁷. Cells were incubated a 30°C (shaking @ 160 rpm) and grown to a final yield of 10⁹ cells mL⁻¹.'

10. In the M&M section, abbreviations such as RH, PPF, ... can be difficult for non-specialists.

Thank you. We have amended these and several others.

Reviewer #3 (Comments for the Author):

This manuscript outlines implementation of an experimental/informatics method to examine the capture and identification of extracellular proteins (termed metaexoproteomics or MEP) synthesized in the rhizosphere of Brassica spp. Their MEP results revealed a significant shift in the metabolically active fraction of the soil microbiota responding to the presence of B. napus roots that was not apparent in the composition of the total microbial community. The authors contend that this metaproteomic assessment of the 'active' plant microbiome at the field-scale demonstrates the importance of moving past a genomic assessment of the plant microbiome in order to determine ecologically important plant-microbe interactions underpinning plant health.

In general, this study is well-designed, systematic, and relies on state-of-the-art approaches in most cases. As is typical in most omics studies, a large amount of data is generated, much of which is somewhat intractable to mine. Nevertheless, the authors have done a nice job in focusing on the extracellular proteins in these natural systems. The manuscript is clearly written, logical in design, and appears scientifically sound. Some specific comments are listed below:

1. I would recommend that the authors consider revising the title a bit to make it more engaging to the lay reader. The current version is a bit bland - maybe add something that provides a bit of "science nugget" about what was learned here.

Thank you for this. We have changed the title to:

'Stimulation of distinct rhizosphere bacteria drive phosphorus and nitrogen mineralisation in oilseed rape under field conditions'

2. Page 2, line 30 - might be good to clarify here if these 1885 proteins are exclusively microbial or also include plant ones?

Agreed, we have amended as follows:

'By combining metagenomics with MEP, 1885 plant, insect and microbial proteins were identified.....'

3. Page 3 - here or elsewhere, it might be important to clarify why focus on "extracellular proteins" rather than the cytosolic ones? Why not examine the entire microbes rather than only their extracellular proteins? Wouldn't this give a more complete glimpse of the entire protein machinery (i.e. functionality) of the system?

We focus on the extracellular proteins as that is where some of the best information regarding ecological interactions can be found, i.e. nutrient acquisition. We tried to circumnavigate the fact that whole proteomes can often be swamped with housekeeping and core metabolic proteins. Whilst this information is good for understanding active microbes, it doesn't give good information about which nutrients are being targeted, and/or which nutrient stress responses are occurring. The vast majority of this information can be retrieved from the extracellular proteome. This is especially true for P and N cycling proteins.

We have added the following sentence to the abstract:

'Information on important ecological interactions being performed by microbes can be investigated by analysing the extracellular protein fraction.'

We have also tweaked the sentence in the introduction:

'This observation is evident in our previous laboratory-based studies investigating individual bacterial responses to phosphate limitation^{27, 28}.'

In addition, we further discuss this in the last section of our introduction.

4. Page 5, line 91 - it might be worth elaborating here that microbial complexity in soils is usually greater than any other environments and leads to considerable problems in metagenome sequencing and assembly, which are critical for quality metaproteome measurements.

Agreed, and we like the way you have succinctly phrased this statement. We have amended as follows:

'Furthermore, microbial complexity in soils is usually greater than any other environment^{1,11,24}, leading to considerable problems in metagenome sequencing and assembly, which are critical for quality metaproteome measurements.

5. Page 6, line 116 - would there be an expectation to see extracellular proteins that would reveal information about microbial "stress" or signalling, in addition to response to elevated carbon? Would this information naturally fall-out of the measurements or would that have to be mined specifically?

As long as 'stress' proteins can be found in the periplasmic, outer membrane or extracellular milieu, this method would detect them. Indeed, we have primarily used this method in previous studies to detect both N and P stress response proteins in both *Flavobacterium* and *Pseudomonas*.

Examples:

Lidbury, I. D. E. A., Murphy, A. R. J., Scanlan, D. J., Bending, G. D., Jones, A. M. E., Moore, J. D., Goodall, A., Hammond, J. P., Wellington, E. M. H., Comparative genomic, proteomic and exoproteomic analyses of three *Pseudomonas* strains reveals novel insights into the

phosphorus scavenging capabilities of soil bacteria. *Environmental Microbiology* **2016**, *18* (10), 3535-3549.

Lidbury, I. D. E. A., Borsetto, C., Murphy, A. R. J., Bottrill, A., Jones, A. M. E., Bending, G. D., Hammond, J. P., Chen, Y., Wellington, E. M. H., Scanlan, D. J., Niche-adaptation in plant-associated *Bacteroidetes* favours specialisation in organic phosphorus mineralisation. *ISME J* **2021**, *15* (4), 1040-1055.

Murphy, A., Transporter characterisation reveals aminoethylphosphonate mineralisation as a key step in the marine phosphorus redox cycle. *Nature Communications* **2021**, *12*, 4554.

6. Page 6, line 131 - the native soil undoubtedly contains a natural microbial community. What is known about this and how might this respond to the media addition and/or the addition of the *P. putida*?

A good question. As we were primarily interested in capturing proteins from *P. putida* BIRD-1 to validate our method, we were not specifically interested in the semi-native community that was enriched in the pots. We do indeed detect peptides/proteins from other *Pseudomonas* and other rhizobacteria. Please see our response to comment 11 below for more information on this.

7. Line 158 - how do the authors know that the 1 hour incubation at room temperature (without the plant) does not change the metabolic activities of the microbes still present in the soil/buffer solution and thus possibly confound the effects?

This is a fair question. This step in the protocol was previously developed by the Wellington group, the product of which was published in 2014 (Rollings-Johnson et al., 2014). There was a significant amount of preliminary work investigating the optimal conditions and buffers to use for the most efficient extraction without impairing the integrity of the sample.

Reference:

Johnson-Rollings, A. S., Wright, H., Masciandaro, G., Macci, C., Doni, S., Calvo-Bado, L. A., Slade, S. E., Vallin Plou, C., Wellington, E. M. H., Exploring the functional soil-microbe interface and exoenzymes through soil metaexoproteomics. *ISME J* **2014**, *8* (10), 2148-2150.

8. Line 181 - would be helpful to give a brief description and defense for ORF clustering at 90% - what resolution is lost.

Agreed. We have added the following text to the methods.

'A 90% clustering value was chosen based on a preliminary database search using a subset of the total MG (0.931 M ORFs), focusing on P cycling and other extracellular proteins (PSet). This reduced database was clustered at both 99% and 90%, and whilst some resolution was lost at 90% clustering, the vast majority of protein clusters were identified. Therefore 90% clustering was applied to constrain the total MG database.'

We used 90% to reduce the size of the database and the size of certain protein clusters related to conserved protein sequences. However, as with all these thresholds, there is a trade off in terms of identifying specific isoforms. We believe that our overall approach has struck a decent compromise.

9. Line 187 - an FDR of 10% seems unacceptably high. This would be unusual, so the authors need to provide more detailed rationale and defence for this choice. I think most readers would find this too liberal, suggesting that many of the IDs may be suspect.

Although an FDR of 10% would certainly be unacceptably high for single strain proteomics, FDR estimation and threshold setting is a complicated issue when applied to complex communities in metaproteomics, as elegantly discussed in the review by Heyer et al (2017). For example, database size can affect FDRs and over-stringent criteria that are used in single strain proteomics leads to high false negative identifications. Prior to our two-step searches, we performed two independent searches using two separate databases. The first, named Pset, contained a comprehensive subset of ORFs (0.931 M) related to P cycling and other extracellular proteins, clustered at 99% similarity. N.B. The comparative datasets showing the difference between 5% protein FDR and 10% protein FDR threshold have been added to the supplementary tables file (Tables S4 and S5).

We acknowledge that these searches are not perfect, but we have aimed to get a balance between reasonable FDR and reduce the number of false negative protein identifications. After comprehensive analysis of the datasets, many of the additional identified peptides and corresponding proteins related to low abundance proteins, and importantly, these made reasonable biological sense, i.e., the pattern of rhizosphere v bulk specialists was perfectly maintained. We found no evidence for any significantly noise in biological inference, hence we accepted the 10% FDR.

Based on our findings, there are slight nuances in interpretation of the data, which are dependent on various combinations of thresholds and database searches, but we found excellent congruence behind the major biological signals illustrated by each. We are very much hoping that the Metaproteomics community, particularly those affiliated with the Metaproteomics Initiative (<https://metaproteomics.org>) will use these datasets to compare and contrast various methods for analysis.

We have added some text to the methods, summarising this, including addition of the preliminary Pset datasets, which have also been uploaded to PRIDE.

We have also added the following to the discussion:

‘Removal of proteins with only one observed peptide, only identified by modified peptides, and allowing for a peptide threshold FDR of 5% and a protein threshold FDR of 10% resulted in a final protein detection of 1895 protein groups (10 contaminants). Setting FDR thresholds is a complex issue when applying these to metaproteomics³⁴. After manual scrutiny (visualised using Scaffold Viewer v4.8.7) of search results obtained using the Pset database and setting the FDR at either 5% (Table S2a) or 10% (Table S2b), we decided on using a relaxed FDR of 10%. The extra peptides/proteins identified by relaxing the FDR

threshold did not occur in a random fashion, suggesting that true biological inference was maintained in the vast majority of cases. Indeed, most extra identified peptides were assigned to proteins already identified using a more stringent FDR, suggesting that isoforms and microdiversity of proteins at the amino acid level restricted the number of peptides/proteins identified. This is perhaps a consequence of our clustering step, but it should be noted that the overall biological inference remains remarkably similar for either chosen FDR threshold (Tables S2a & S2b).'

Reference:

Heyer R, Schallert K, Zoun R, Becher B, Saake G, Benndorf D. Challenges and perspectives of metaproteomic data analysis. *J Biotechnol.* 2017; 261:24-36. doi: 10.1016/j.jbiotec.2017.06.1201.

10. Line 245 - how does this taxonomic profiling compare to the attempt to assemble MAGs from metagenome data? Did the authors consider MAG assembly, even though it would be variable across the range of taxa?

We did indeed try and assemble MAGs from our contigs, using the method developed by the Quince group. Unfortunately, almost all MAGs were very poor quality (based on SCG occurrence within a given MAG). These were either incomplete or contaminated. This is why we went with using the SCGs themselves.

11. Line 278 - was the sand:soil mix sterilized to kill endogenous microbes? If not, how would this affect the measurement? Should the searching have been done with a larger "pseudo-metagenome" to provide IDs for other members? What is the possibility of redundant peptides/proteins from other microbes in the endogenous community?

We have now included a supplementary table (new table S1c) which is the result of a similar Maxquant search using a *P. putida* database with additional *Pseudomonas* and non-*Pseudomonas* extracellular proteins (phosphatases, substrate binding proteins, pectinases, etc.) extracted from bacterial genomes deposited in the IMG/JGI database. Indeed, we do find some evidence of a 'semi-native' microbiome in our pots as well as the presence of proteins related to other plant-associated *Pseudomonas* spp. However, we still identify *P. putida* specific proteins within our sample suggesting that we are retrieving *bona fide* *P. putida* hits. Importantly, this extra search further demonstrates the ability of our extraction method for capturing proteins in the rhizosphere, even for the native microbiome. We have added the following section to acknowledge this point/concern.

'We also performed a third search using the *P. putida* BIRD-1 proteome and additional protein sequences related to various extracellular proteins (substrate binding proteins, phosphatases, chitinases, pectinases, etc.) retrieved from bacterial genomes deposited in the IMG/JGI database (accessed on 05/06/2018). This search indicated that specific *P. putida* BIRD-1 proteins were still retrieved, in addition to those from closely related *Pseudomonas* strains, and exoproteins produced by taxonomically divergent rhizobacteria (Table S1c).'

12. Line 334 - how close are these genomes - species, strains?

As per the other reviewer's comments. We have tweaked this section. These strains represent four distinct *Pseudomonas* sub-lineages, which are frequently found in the rhizosphere. We note that the original DIAMOND annotations do indeed reveal the presence of multiple strains, but with comprehensive MAGs, we are cautious about stating any more than the fact there appear to be multiple strains active.

13. Line 353 - the presence of intracellular proteins, while not surprising, suggested that normal cellular lysis is occurring. How do the authors sort out which proteins are definitively "extracellular" from those that simply result from cell death/lysis?

Line 363? Extracellular proteins were determined using various online tools, such as SecretomeP, LipoP, etc. This was especially important for proteins with unknown function, annotated as either 'hypothetical' or 'uncharacterised'. However, whilst not discussed in detail, there are several papers showing that secretion of certain cytoplasmic proteins occurs and is known as 'moonlighting'. Thus, we try and stay clear of these potential phenomena, and try to focus on how the method can help enrich extracellular proteins.

We have added a sentence to the methods clarifying how we screened for predicted extracellular proteins, line 211:

'In silico protein sorting prediction tools, such as secretomeP 2.0, LipoP, and SignalP (<https://services.healthtech.dtu.dk/>) were used to determine if proteins were truly extracellular or simply intracellular and released during cell lysis/death.'

Reference:

Jeffery Constance J. (2018) Protein moonlighting: what is it, and why is it important? *Phil. Trans. R. Soc. B* 373:20160523-20160523.

14. Line 438 - might be worthwhile pointing out here that while meta-transcriptomic studies also provide "functional information about the microbiome," they lack the ability to spatially resolve intracellular vs. extracellular activities.

Thank you, this is an excellent point. We have added the following sentence:

'In addition, whilst metatranscriptomics can provide information on activity, it suffers from neglecting post-transcriptional and post-translational regulation of protein synthesis, as well as an inability to spatially resolve protein compartmentalisation, i.e. intra- versus extra-cellular location.'

15. Line 453 - do the authors feel that their systems are relatively stable and thus a single time-point measurement is likely to reflect sustained activity vs. temporal "blips?"

We acknowledge this and have added a sentence at the end of this section highlighting the need for more refined temporal sampling.

'Further investigation should now focus on determining how the active fraction of the community changes over time, particularly at different growth stages. This would help determine the stability of field-grown plant microbiomes, which could ultimately be used to determine early signs of pathogen-induced dysbiosis.'

16. Line 741 - the use of NSAF represents an older version of relative quantification. This is not to say that it is incorrect but more recent methods to exploit extracted peak areas are more tractable with high resolution LC separations coupled to high mass accuracy MS systems. These also afford deeper dynamic range. The authors should address their choice of NSAF over peak areas?

We fully acknowledge this quantification method is now relatively 'old'. Indeed, for our field grown samples, we used peak areas to quantify. The reason we maintained NSAF values was to easily compare our data with our pre-existing dataset, obtained from in vitro cells. The published data in this paper uses NSAF. Therefore, to make it easier for the reader to cross reference with our older paper, we thought it most appropriate to use the same values here. We have provided access to all spectral files to enable readers to re-analyse using peak areas, or indeed alternative NSAF algorithms too.

We have amended the sentence starting line 290, to make this point:

'Whilst normalised spectral abundance factor (NSAF) quantification methods have now been superseded by various peak areas methods, we used NSAF here to make easier comparisons with our previously published datasets²⁷.

We have also stated LFQ was used to determine relative abundances in the field MEP samples, line 324.

'The relative abundance of proteins was quantified using label-free quantification (LFQ) values. There was a significant difference (Adonis: $R^2=0.736$, $P = 0.001$) in the proteomic profiles of bulk and rhizosphere samples (Fig. 2C).'

June 12, 2022

Dr. Ian Dennis Edmund Alan Lidbury
The University of Sheffield
Department of Animal and Plant Science
Sheffield
United Kingdom

Re: mSystems00025-22R1 (Stimulation of distinct rhizosphere bacteria drive phosphorus and nitrogen mineralisation in oilseed rape under field conditions)

Dear Dr. Ian Dennis Edmund Alan Lidbury:

Thank you for being so thorough in addressing the reviewer comments. This is an exciting piece of work, and the paper will undoubtedly be impactful in the field of metaproteomics.

Your manuscript has been accepted, and I am forwarding it to the ASM Journals Department for publication. For your reference, ASM Journals' address is given below. Before it can be scheduled for publication, your manuscript will be checked by the mSystems production staff to make sure that all elements meet the technical requirements for publication. They will contact you if anything needs to be revised before copyediting and production can begin. Otherwise, you will be notified when your proofs are ready to be viewed.

Publication Fees:

We recognize that the video files can become quite large, and so to avoid quality loss ASM suggests sending the video file via <https://www.wetransfer.com/>. When you have a final version of the video and the still ready to share, please send it to mSystems staff at mSystems@asmusa.org.

For mSystems research articles, if you would like to submit an image for consideration as the Featured Image for an issue, please contact mSystems staff at mSystems@asmusa.org.

Sincerely,

Michelle Heck
Editor, mSystems

Journals Department
Supplemental Material 1: Accept
Supplemental Material 6: Accept
Supplemental Material 5: Accept
Supplemental Material 2: Accept
Supplemental Material 4: Accept
Supplemental Material 7: Accept
Supplemental Material 8: Accept
Supplemental Material 9: Accept
Supplemental Material 3: Accept